# CTRP3 and serum triglycerides in children aged 7-10 years

Arsham Alamian[1], Jo-Ann Marrs[2], W. Andrew Clark[3], Kristy L. Thomas[4], Jonathan M. Peterson[4,5] *

1 School of Nursing and Health Studies, University of Miami, Coral Gables, Florida, United States of America, 2 College of Nursing, East Tennessee State University, Johnson City, Tennessee, United States of America, 3 College of Clinical and Rehabilitative Health Sciences, East Tennessee State University, Johnson City, Tennessee, United States of America, 4 Department of Biomedical Sciences, Quillen College of Medicine, East Tennessee State University, Johnson City, Tennessee, United States of America, 5 Department of Health Sciences, College of Public Health, East Tennessee State University, Johnson City, Tennessee, United States of America

* petersonjm1@etsu.edu

## Abstract

### Introduction

The prevalence of obesity-related disorders has been steadily increasing over the past couple of decades. Diseases that were once only detected in adults are now prevalent in children, such as hyperlipidemia. The adipose tissue-derived hormonal factor C1q TNF Related Protein 3 (CTRP3) has been linked to triglyceride regulation especially in animal models. However, the relationship between circulating CTRP3 levels and obesity-related disorders in human subjects is controversial. CTRP3 can circulate in different oligomeric complexes: trimeric (<100 kDa), middle molecular weight (100–300 kDa), and high molecular weight (HMW) oligomeric complexes (>300 kDa). Previous work has identified that it is not the total amount of CTRP3 present in the serum, but the specific circulating oligomeric complexes that appear to be indicative of the relationship between CTRP3 and serum lipids levels. However, this work has not been examined in children. Therefore, the purpose of this study was to compare the levels of different oligomeric complexes of CTRP3 and circulating lipid levels among young children (aged 7–10 years).

### Methods

Morphometric data and serum samples were collected and analyzed from a cross-sectional population of 62 children of self-identified Hispanic origin from a community health center, between 2015 and 2016. Serum analysis included adiponectin, insulin, leptin, ghrelin, glucagon, C-reactive peptide, triglyceride, cholesterol, IL-6, TNF, and CTRP3. Correlation analyses were conducted to explore the relationships between CTRP3 and other biomarkers.

### Results

Total CTRP3 concentrations were significantly positively correlated with total cholesterol and HDL cholesterol. Whereas, HMW CTRP3 was not significantly associated with any

**Data Availability Statement:** All relevant data are within the manuscript and its Supporting information files.

**Funding:** This work was supported by grants from the Tennessee Board of Regents [Diversity

Research Grant TBR E210029] and the National Institute of Diabetes and Digestive and Kidney Diseases [R15 DK114740-01A1]. The funders had no role in study design, data collection and analysis, decision to publish, or preparation of the manuscript.

**Competing interests:** The authors have declared that no competing interests exist.

variable measured. Conversely, the middle molecular weight (MMW) CTRP3 was negatively correlated with triglycerides levels, and very low-density lipoprotein (VLDL), insulin, and body mass index (BMI). The negative correlations between MMW CTRP3 and triglycerides and VLDLs were particularly strong ($r^2$ = -0.826 and -0.827, respectively).

## Conclusion

Overall, these data indicate that the circulating oligomeric state of CTRP3 and not just total CTRP3 level is important for understanding the association between CTRP3 and metabolic diseases. Further, this work indicates that MMW CTRP3 plays an important role in triglyceride and VLDL regulation which requires further study.

## Introduction

Childhood obesity is a growing epidemic in the Unites States; its prevalence more than doubled in the past 30 years from ~7% in 1980 to 17.7% in 2012 in 6- to 11-year-olds [1, 2]. Obese children are at a significantly higher risk for developing many types of cancers, cardiovascular disease, diabetes and other metabolic disorders [3, 4]. However, there is a paucity of data regarding the development of metabolic dysfunction in children, especially among the Hispanic population. Childhood obesity is influenced by a variety of environmental, dietary, and genetic factors, many of which are actively being investigated. Our lab focuses on understanding the influence of adipose tissue health and specifically adipose tissue-derived secreted hormonal factors (hereafter referred to as adipokines) on the development of obesity and metabolic disorders. This manuscript focuses specifically on associations of C1q TNF Related Protein 3 (CTRP3) on obesity-related metabolic parameters.

Adipose tissue secretes many bioactive molecules that circulate in blood, collectively termed adipokines [5–12]. To date, over 70 adipokines have been identified [8, 13]. Leptin is the most well-known and highly studied adipokine and disruptions in proper leptin function result in severe obesity through hyperphagia (overeating) and associated metabolic disorders [14]. The second most widely studied adipokine is adiponectin. Adiponectin modulates a number of metabolic processes, including glucose regulation and fatty acid oxidation [15]. Obesity is associated with lower adiponectin levels in adults [16], and hypoadiponectinemia is a consequence of the development of obesity in childhood [17, 18]. Additionally, a novel family of secreted humoral factors, C1q TNF Related Proteins, (abbreviated CTRP1 through 15) have been identified based upon homology to adiponectin [9, 11, 19]. Reflecting profound biological potency, the initial characterization of these adipose tissue-derived CTRP factors finds wide-ranging effects upon metabolism, inflammation, and cell-growth in multiple tissue types. The associations between these factors and human health has only begun to be explored in adult populations and this manuscript is the first to explore any of these factors in children.

In animal models CTRP3 has been shown to improve insulin sensitivity as well as inhibit the development of both non-alcoholic fatty liver disease and alcoholic fatty liver disease [20–23]. However, there have been conflicting studies regarding the relationship between circulating CTRP3 levels and metabolic status. For example, CTRP3 has been reported to be elevated [24], not changed [25, 26], or reduced with obesity [24, 27–30]. These conflicting data suggest that measuring total circulating levels of CTRP3 is insufficient for determining the role of CTRP3 in human health.

Although CTRP3 is an approximately 26 kDa protein, when secreted CTRP3 forms higher order molecule structures such as a trimer (~78–90 kDa), a six-nine subunit oligomer also known as a middle molecular weight oligomer (MMW, ~180–270 kDa), or as a high molecular weight oligomer (HMW, >300 kDa) [9]. Our previous work demonstrated that CTRP3 is only found in human circulation as MMW or HMW isoforms [26]; this was also confirmed experimentally in the pediatric study population presented herein (S1 Fig). In addition, Trogen et al. [26] found that the differences in the circulating oligomeric forms of CTRP3 had strong correlations with obesity and other metabolic variables, especially circulating triglyceride levels. Therefore, we hypothesized that the MMW and HMW oligometric CTRP3 will be correlated with metabolic variables, specifically circulating serum triglyceride levels. To test this hypothesis we re-examined a sample collected from a population of Hispanic children who we have previously studied shown to be at increased risk for obesity and metabolic diseases [31].

## Methods

### Study design and sample

Data for this study came from a previously performed cross-sectional study of metabolic syndrome in pre-adolescent Hispanic Children, receiving well-child care at a community health center, from June 2015 to September 2016 [31]. The study was conducted by the APPalachian Obesity and METabolic diseases (APPOMET) Working Group, an interdisciplinary group of researchers including epidemiologists, nutritionists, nurses and basic science researchers. The study was reviewed and approved by the Institutional Review Board at East Tennessee State (IRB#: 0414.16s).

Inclusion criteria for this study were: being 7–10 years of age; being of Hispanic origin by self-identification; and not having a serious physical or mental illness. Parents were provided written information about the study in either Spanish or English Language. Parents understood that participation in the study was voluntary and they received assurance of the confidentiality of the data which they provided. Parental written consent and a child written assent were obtained before data collection. A total of 62 children aged 7–10 years old were included in this study.

### Measurements

As described previously a pediatric nurse practitioner measured children's height, weight, waist circumference and blood pressure using standard protocols [31]. A laboratory technician drew four milliliters (4mls) of blood from the ante-cubital fossa of each child into a serum separator tube (SST) and a ethylenediamine tetra-acetic acid (EDTA) tube. The blood samples were stored at -80˚C until analysis.

Blood sample analysis: Total Cholesterol, LDL, HDL, C Reactive protein, Triglycerides, and VLDL, were performed by ETSU Clinical Laboratory: an accredited reference lab (Center for Medicare & Medicaid Services Clinical Laboratory, certification number 44D0659180). Adiponectin, IL-6, C-peptide, Ghrelin, Glucagon, and Leptin analysis were performed on Bio-Rad Bio-Plex Mag-Pix with commercially available assays according to manufactures instructions (Bio-Rad Bio-Plex, Catalog numbers: 171A7002M; 171-A7001M; & #171-AA001M).

CTRP3 Analysis: Serum samples were diluted to 1:40 with phosphate-buffered saline containing protease inhibitor cocktail (Bimake, Catalog numbers #B14001) and were separated by size using centrifugal separation (Sartorius™ Vivaspin™; VS0151) according to manufactures' directions. The flow through contains all molecules below 300 kDa in size and the concentrate contains only those molecules equal to or above 300 kDa in size. Total (unseparated samples), high molecular weight isomer (HMW), and middle molecular weight isomer (MMW) of

CTRP3 were measured using the CTRP3 ELISA (R&D systems, Catalog number #DY7925-05).

## Statistical analysis

Descriptive statistics (mean and standard deviation) were calculated for all measured variables. An unpaired two-tailed t test was used to compare means between males and females for all variables tested; no sex specific differences were identified, and all samples were combined for further analysis. Normality was tested with a D'Agostino & Pearson omnibus normality test, and many of the variables were found to be not normally distributed. Therefore, all correlation coefficients were calculated via the Spearman's rank-order correlation test. All statistical analyses were performed by Graphpad Prism 6.

# Results

## Demographic data

The study included 62 children (29 males and 33 females) with an average BMI of 20.2 ± 4.9 kg/m$^2$ (calculated using the 2000 CDC growth charts) [32], and an average age of 8.7 ± 1.1 years. There were no significant sex differences in any variable measured as determined by an unpaired two-tailed t test.

## Subject characteristics

The mean, standard deviation, min and max values are reported for all serum values in Table 1.

**Table 1. The mean, standard deviation, min and max values are reported for all serum values (n = 62).**

|  | *Mean ± SD* | *Range (min–max)* |
|---|---|---|
| *Total CTRP3 (ng/mL)* | 103.2 ± 22.6 | 38.9–167.6 |
| *MMW CTRP3 (ng/mL)* | 29.7 ± 20.0 | 5.7–84.6 |
| *HMW CTRP3 (ng/mL)* | 89.4 ± 26.7 | 30.6–156.4 |
| *Adiponectin (ug/mL)* | 24.0 ± 13.5 | 2.2–73.1 |
| *C-Peptide (pg/mL)* | 1243 ± 744 | 277–3772 |
| *Ghrelin (pg/mL)* | 232.1 ± 26.2 | 7.1–1338 |
| *Glucagon (pg/mL)* | 160.0 ± 17.8 | 34.2–751.8 |
| *Leptin (pg/mL)* | 6360 ± 991 | 174–33815 |
| *IL-6 (pg/mL)* | 2.7 ± 5.2 | 0.05–33.6 |
| *TNF (pg/mL)* | 8.1 ± 18.5 | 0.58–148.3 |
| *C-Reactive Protein (pg/mL)* | 2.7 ±3.8 | 0.21–18.8 |
| *Insulin (pg/mL)* | 18.2 ± 18.4 | 2.0–114.0 |
| *Triglycerides (mg/dL)* | 111.3 ± 48.9 | 35.0–263.0 |
| *Total Cholesterol (mg/dL)* | 148 ± 24 | 75–195 |
| *HDL (mg/dL)* | 49.1 ± 10.9 | 27.3–77.3 |
| *LDL (mg/dL)* | 77.0 ± 22.3 | 26.6–119.6 |
| *VLDL (mg/dL)* | 22.5 ± 9.7 | 7.0–53.0 |

Abbreviations: MMW, middle molecular weight; HMW, high molecule weight; IL-6, Interleukin 6; TNF, tumor necrosis factor; HDL, high-density lipoproteins; LDL, low-density lipoproteins; VLDL, very low density lipoprotein.

## CTRP3 correlations

Total CTRP3 concentrations were significantly positively correlated with total cholesterol
(p = 0.034, $r^2$ = 0.270), and HDL cholesterol (p = 0.033, $r^2$ = 0.271) (S1 Table). The relationship
between total CTRP3 and adiponectin approached significance (p = 0.06). HMW CTRP3 was
not significantly associated with any other variable measured, however the positive relationship between HMW CTRP3 and HDL cholesterol levels did approach significance (p = 0.095)
(S2 Table). On the other hand, MMW CTRP3 was significantly and negatively correlated with
insulin, triglycerides, VLDL cholesterol and BMI (data and p-values shown in Fig 1 and S3
Table). MMW CTRP3 was found to be positively correlated with adiponectin (p = 0.039, $r^2$ =
0.262). Also, it is worth noting that the positive correlation between MMW CTRP3 and ghrelin
and HDL cholesterol both approached significance (p = 0.09 and p = 0.07 respectively).

## Discussion

Dyslipidemia, the abnormal amount of lipids such as triglyceride and cholesterol, is commonly
induced by obesity and is a leading contributor to the development of metabolic syndrome.
Metabolic syndrome is a significant public health concern and the most prevalent cause of cardiovascular disease and type 2 diabetes. There is growing evidence that children and adolescents are increasingly affected by obesity and metabolic syndrome [1–3, 33–35]. However,
there is a paucity of information on the relationship of emerging hormones and their relationship with obesity related disorders, especially in young children. CTRP3 is an adipokine that
has been shown to improve dyslipidemia, prevent ectopic lipid accumulation, and reduce the
impact of cardiovascular disease [19–22, 24–27, 36, 37]. However, the study of the relationship
between circulating CTRP3 levels and human health outcomes and characteristics has produced conflicting data. Specifically, total circulating CTRP3 levels are reported to be either
reduced [27, 29, 37, 38] or elevated [39] depending on the study population. However, it has
been speculated that the oligomer conformation of CTRP3 is important for its functional activity [19, 22, 40, 41], and this has been rarely studied. To date only one other study has examined
the oligomeric status of CTRP3 with human disease [26]. Trogen et al. (2019) determined that
all circulating CTRP3 in humans is found as either the MMW or HMW oligomer, with the trimer and monomer of CTRP3 being undetectable. Further, the authors found that the oligomeric state of CTRP3 was significantly correlated to serum triglyceride levels, but not diabetic
or obesity status [21].

 This is the first study to examine the circulating oligomeric state of CTRP3 in a pediatric
population (7–10 years of age). The major significant finding of this study is that the oligomeric status of CTRP3 is strongly negatively correlated with circulating triglycerides and
VLDL cholesterol levels. Further, we identified that the circulating oligomeric state, specifically
the MMW CTRP3, and not total CTRP3 concentration is negatively associated with BMI and
insulin levels. It has been hypothesized that the HMW CTRP3 isomer is the inactive form and
that CTRP3 must undergo some form of cleavage to become active [19, 22, 42]. Our data supports the hypothesis, as although the HMW CTRP3 isomer was the predominate form of
CTRP3 found in the blood, the concentration of MMW CTRP3 was the oligomeric state that
was negatively correlated with triglyceride and VLDL cholesterol levels. Interestingly, this
study did not identify a relationship between HMW CTRP3 and triglyceride levels as was identified by Trogen el al [32]. However, the discrepancy between the two studies are potentially
due to the different population as Trogen et al. examined CTRP3 oligomers in an obese adult
population (average BMI >30 kg/m2) with half of the population studied suffering from type 2
diabetes. Further, the negative correlation between the MMW oligomer of CTRP3 and serum
triglycerides in the study described by Trogen et al. [32] approached significance (p = 0.06),

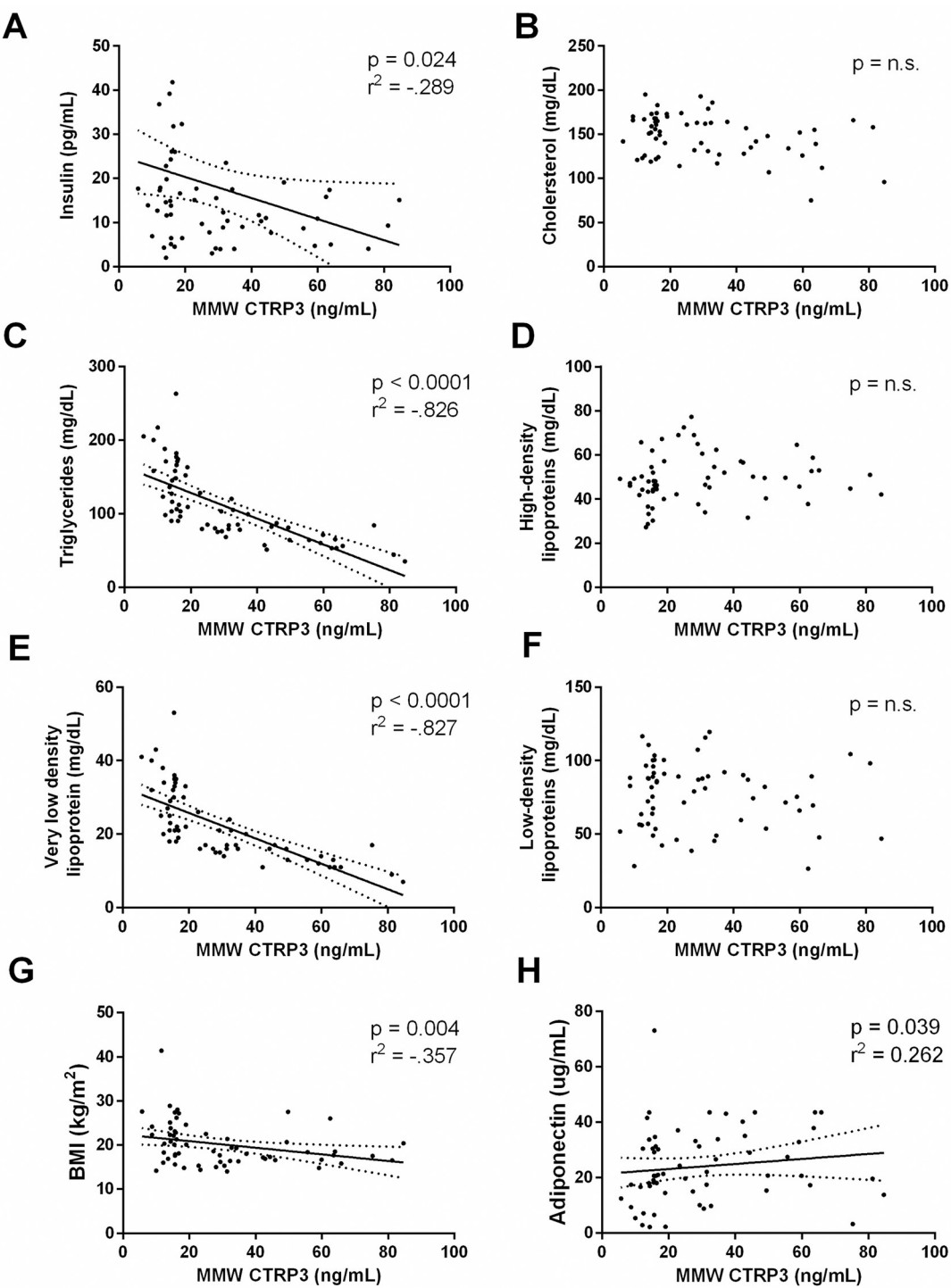

**Fig 1. Correlations between MMW CTRP3 and serum analytes.** The Spearman's rank-order correlation coefficient was calculated between MMW CTRP3 and serum analytes. Each dot represents a unique individual. When a significant correlation exists the linear relationship is graphed with the solid line and the 95% confidence interval with the dotted lines. Abbreviations: n.s. = not significant, MMW CTRP3 = medium molecular weight CTRP3 oligomer (<300 kDa).

which agrees with the current findings. Together these data indicate that the oligomeric state is important to the role of CTRP3 and needs to be studied in additional healthy and diseased populations.

## Conclusions

Understanding how circulating novel hormonal factors contribute to the development of disease is an essential step towards developing intervention strategies to treat/prevent childhood obesity and metabolic syndrome. Our finding that the specific oligomeric conformation of CTRP3 was strongly and negatively correlated with dyslipidemia indicates that CTRP3 plays a key role in metabolic health and its dysregulation can be an early sign of the development of metabolic disease.

## Study limitations

The relatively small sample size (n = 62) limits some of the significance of the study findings. However, this is the second study with a unique population which has identified the correlation of the oligomeric conformation of CTRP3 and triglyceride levels. Further, the strong negative correlation identified within this population between MMW CTRP3 and triglyceride as well as VLDL cholesterol levels, combined with the previously published work in animals, strongly supports the role of CTRP3 in regulating triglycerides levels.

The cross-sectional nature of this study precludes making causal claims. However, combined with the previous findings of CTRP3 preventing hepatic triglyceride synthesis in rodents [20, 21], a potential mechanism for CTRP3-induced triglyceride regulation has been identified and requires further analysis. Lastly, we did not study the oligomeric state of other adipokines, especially adiponectin, as it was outside the scope and budget of the current study. However, the relationship between the oligomerization state of the other pertinent adipokines requires further study in the pediatric population.

## Supporting information

**S1 Data.**
(XLSX)

**S1 Fig. CTRP3 was not detectable in the LMW fraction.**
(DOCX)

**S1 Table. Spearman's rank-order correlation coefficient for total CTRP3 and other metabolic parameters.**
(DOCX)

**S2 Table. Spearman's rank-order correlation coefficient for HMW CTRP3 and other metabolic parameters.**
(DOCX)

**S3 Table. Spearman's rank-order correlation coefficient for MMW CTRP3 and other metabolic parameters.**
(DOCX)

## Author Contributions

**Conceptualization:** Arsham Alamian, Jonathan M. Peterson.

**Data curation:** Jo-Ann Marrs.

**Formal analysis:** Arsham Alamian, Jonathan M. Peterson.

**Funding acquisition:** W. Andrew Clark, Jonathan M. Peterson.

**Investigation:** Jo-Ann Marrs, Kristy L. Thomas.

**Methodology:** W. Andrew Clark.

**Project administration:** W. Andrew Clark.

**Resources:** Jonathan M. Peterson.

**Writing – original draft:** Jonathan M. Peterson.

**Writing – review & editing:** Arsham Alamian, Jo-Ann Marrs, W. Andrew Clark, Kristy L. Thomas, Jonathan M. Peterson.

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
