## [Decision Letter · Decision Letter 0]

15 Sep 2020

PONE-D-20-22639

CTRP3 and serum triglycerides in children aged 7-10 years

PLOS ONE

Dear Dr. Peterson,

Thank you for submitting your manuscript to PLOS ONE. After careful consideration, we feel that it has merit but does not fully meet PLOS ONE’s publication criteria as it currently stands. Therefore, we invite you to submit a revised version of the manuscript that addresses the points raised during the review process.

Please address properly all reviewer's comments.

We look forward to receiving your revised manuscript.

Kind regards,

Paolo Magni

Academic Editor

PLOS ONE

Journal Requirements:

3.Thank you for stating the following in the Funding Section of your manuscript:

[This work was supported by grants from the Tennessee Board of Regents [Diversity

Research Grant TBR (2014-2015)] and the National Institute of Diabetes and Digestive

and Kidney Diseases [R15 DK114740-01A1]. The funders had no role in study design,

data collection and analysis, decision to publish, or preparation of the manuscript.]

 [The funders had no role in study design, data collection and analysis, decision to publish, or preparation of the manuscript.]

4. Please include your tables as part of your main manuscript and remove the individual files. Please note that supplementary tables (should remain/ be uploaded) as separate "supporting information" files

Reviewers' comments:

Reviewer's Responses to Questions

**Comments to the Author**

1. Is the manuscript technically sound, and do the data support the conclusions?

Reviewer #1: Partly

2. Has the statistical analysis been performed appropriately and rigorously? 

Reviewer #1: Yes

3. Have the authors made all data underlying the findings in their manuscript fully available?

Reviewer #1: Yes

4. Is the manuscript presented in an intelligible fashion and written in standard English?

Reviewer #1: Yes

5. Review Comments to the Author

Reviewer #1: Summary: This study measured CTRP3 HMW and presumed MMW circulating levels from blood samples taken from pediatric donors. Key finding was that circulating triglycerides and VLDL levels negatively correlated with MMW CTRP3, while HMW CTRP3 did not show any statistically significant correlation with the measured parameters.

Significance of study: The circulating levels of CTRP3 oligomeric complexes have previously been examined in an adult cohort with or without Type 2 Diabetes Mellitus. This is a similar study on a pediatric cohort. While the reported parameters are similar between the two studies, CTRP3 levels have not previously been examined in a pediatric demographic, and this study provides novel information relevant to the growing epidemic of childhood obesity and type 2 diabetes.

General comments: The manuscript is clearly written except for some typographical errors throughout the manuscript and a few run-on sentences. Authors should carefully review the manuscript for typographical and grammatical errors. Appropriate statistical analyses have been performed on the data presented.

Specific comments:

1) In Line 61, authors incorrectly state that adipose is the “largest endocrine organ by mass”. Muscle, the largest organ by mass, also secretes a vast array of bioactive molecules including myokines and cytokines. This statement should be corrected.

2) The authors use centrifugal separation to differentiate molecules above 300kDa from those under 300kDa, and conclude that CTRP3 measurements in these two fractions correspond to HMW and MMW CTRP3 populations. However, there is the possibility that the <300kDa fraction contains both LMW and MMW species, and the authors do not demonstrate experimentally that MMW CTRP3 is the only CTRP3 oligomeric species in this fraction.

Although the authors cite a previous publication that reported no detectable LMW species in plasma, it should not be assumed that this is the case in a pediatric population. Authors should demonstrate that CTRP3 is not detected (by immunoblot or ELISA) in a <50kDa fraction.

3) Authors should include a schematic or table to summarize the direction of correlations between the measured parameters and HMW or MMW CTRP3 and total CTRP3. This will help the reader better visualize these patterns between the different CTRP3 oligomeric forms.

4) Typographical error in y-axis Fig 1b.

5) Data for HMW CTRP3 correlations with the measured parameters should be provided as a supplement or otherwise, as conclusions are drawn from these data.

6. PLOS authors have the option to publish the peer review history of their article (what does this mean?). If published, this will include your full peer review and any attached files.

Reviewer #1: No

---

## [Author Response · Author response to Decision Letter 0]

6 Oct 2020

Summary: This study measured CTRP3 HMW and presumed MMW circulating levels from blood samples taken from pediatric donors. Key finding was that circulating triglycerides and VLDL levels negatively correlated with MMW CTRP3, while HMW CTRP3 did not show any statistically significant correlation with the measured parameters.

Significance: The circulating levels of CTRP3 oligomeric complexes have previously been examined in an adult cohort with or without Type 2 Diabetes Mellitus. This is a similar study on a pediatric cohort. While the reported parameters are similar between the two studies, CTRP3 levels have not previously been examined in a pediatric demographic, and this study provides novel information relevant to the growing epidemic of childhood obesity and type 2 diabetes.

The manuscript is clearly written except for some typographical errors throughout the manuscript and a few run-on sentences. Authors should carefully review the manuscript for typographical and grammatical errors. Appropriate statistical analyses have been performed on the data presented. 

• We thank the reviewer for support of our manuscript and suggestions for improvement. We would like to apologize for typographical errors and oversights in the previous version and attest that all authors have carefully reviewed the revised version.

Specific comments:

1) In Line 61, authors incorrectly state that adipose is the “largest endocrine organ by mass”. Muscle, the largest organ by mass, also secretes a vast array of bioactive molecules including myokines and cytokines. This statement should be corrected.

• We have removed this statement as the reviewer is correct that in healthy weight persons muscle mass far exceeds adipose tissue. 

2) The authors use centrifugal separation to differentiate molecules above 300kDa from those under 300kDa, and conclude that CTRP3 measurements in these two fractions correspond to HMW and MMW CTRP3 populations. However, there is the possibility that the <300kDa fraction contains both LMW and MMW species, and the authors do not demonstrate experimentally that MMW CTRP3 is the only CTRP3 oligomeric species in this fraction. Although the authors cite a previous publication that reported no detectable LMW species in plasma, it should not be assumed that this is the case in a pediatric population. Authors should demonstrate that CTRP3 is not detected (by immunoblot or ELISA) in a <50kDa fraction.

• This is an excellent point by the reviewer. We performed additional experiments on these samples to attempt to detect CTRP3 in the <100 kDa fraction by immunoblot. We performed the assay multiple times with three different CTRP3 antibodies and we were unable to detect CTRP3 in the less than 100 kDa fraction (had we been able to detect CTRP3 <100 we would have also examined a <50 kDa fraction). We added a representative image as a supplemental file (Supplemental figure 1). 

3) Authors should include a schematic or table to summarize the direction of correlations between the measured parameters and HMW or MMW CTRP3 and total CTRP3. This will help the reader better visualize these patterns between the different CTRP3 oligomeric forms. 

• We appreciate the reviewer’s suggestion and have added the requested information as 3 separate supplemental tables detailing the correlation coefficients between metabolic parameters and total CTRP3 (Table S1), HMW CTRP3 (Table S2), and MMW CTRP3 (Table S3).

---

## [Decision Letter · Decision Letter 1]

21 Oct 2020

CTRP3 and serum triglycerides in children aged 7-10 years

PONE-D-20-22639R1

Dear Dr. Peterson,

We’re pleased to inform you that your manuscript has been judged scientifically suitable for publication and will be formally accepted for publication once it meets all outstanding technical requirements.

Kind regards,

Paolo Magni

Academic Editor

PLOS ONE

Additional Editor Comments (optional):

The comments have all been addressed.

I recommend to follow this suggestion:

It would be good practice to include a loading control for your western blot image in S1 to show that lack of detection of CTRP3 in the <100Kda fractions is not due to total protein not passing through the filtration process. Transferrin (77Kd) is a good loading control.

Reviewers' comments:

Reviewer's Responses to Questions

**Comments to the Author**

1. If the authors have adequately addressed your comments raised in a previous round of review and you feel that this manuscript is now acceptable for publication, you may indicate that here to bypass the “Comments to the Author” section, enter your conflict of interest statement in the “Confidential to Editor” section, and submit your "Accept" recommendation.

Reviewer #1: All comments have been addressed

2. Is the manuscript technically sound, and do the data support the conclusions?

Reviewer #1: (No Response)

3. Has the statistical analysis been performed appropriately and rigorously? 

Reviewer #1: (No Response)

4. Have the authors made all data underlying the findings in their manuscript fully available?

Reviewer #1: (No Response)

5. Is the manuscript presented in an intelligible fashion and written in standard English?

Reviewer #1: Yes

6. Review Comments to the Author

Reviewer #1: Thank you for addressing the comments. Additionally, it would be good practice to include a loading control for your western blot image in S1 to show that lack of detection of CTRP3 in the <100Kda fractions is not due to total protein not passing through the filtration process. Transferrin (77Kd) is a good loading control.

7. PLOS authors have the option to publish the peer review history of their article (what does this mean?). If published, this will include your full peer review and any attached files.

Reviewer #1: No

---

## [Editor Report · Acceptance letter]

13 Nov 2020

PONE-D-20-22639R1 

CTRP3 and serum triglycerides in children aged 7-10 years 

Dear Dr. Peterson:

I'm pleased to inform you that your manuscript has been deemed suitable for publication in PLOS ONE. Congratulations! Your manuscript is now with our production department. 

Kind regards, 

on behalf of

Prof. Paolo Magni 

Academic Editor

PLOS ONE